# Omega-3 Fatty Acid and Vitamin D Supplementations Partially Reversed Metabolic Disorders and Restored Gut Microbiota in Obese Wistar Rats

**DOI:** 10.3390/biology13121070

**Published:** 2024-12-20

**Authors:** Dylan Le Jan, Mohamed Siliman Misha, Sandrine Destrumelle, Olivia Terceve, Chantal Thorin, Thibaut Larcher, Mireille Ledevin, Jean-Claude Desfontis, Eric Betti, Yassine Mallem

**Affiliations:** 1Nutrition, PathoPhysiology and Pharmacology (NP3) Unit, Oniris, 101 Rte de Gachet, 44300 Nantes, France; m.silimanmisha@gmail.com (M.S.M.); sandrine.destrumelle@oniris-nantes.fr (S.D.); olivia.terceve@gmail.com (O.T.); jean-claude.desfontis@oniris-nantes.fr (J.-C.D.); eric.betti@oniris-nantes.fr (E.B.); 2Institut National de Recherche pour l’Agriculture, l’Alimentation et l’Environnement, Oniris, UMR 703, PanTher, APEX, 44307 Nantes, France; chantal.thorin@oniris-nantes.fr (C.T.); thibaut.larcher@oniris-nantes.fr (T.L.); mireille.ledevin@oniris-nantes.fr (M.L.)

**Keywords:** obesity, vitamin D, omega-3 fatty acids, metabolic disorders, intestinal microbiota

## Abstract

Obesity is a major health issue affecting both humans and animals, leading to various related health problems. This study explored whether vitamin D and omega-3 fatty acids, alone or together, could help manage obesity and its related disorders in rats. Over the course of 26 weeks, rats were first fed a diet high in fat and sugar to induce obesity. They were then given either vitamin D, omega-3 fatty acids, both together, or no supplement at all. The results showed that vitamin D helped improve their blood sugar levels and reduced liver damage, while omega-3 fatty acids slowed weight gain, reduced fat, and improved gut health. When used together, these supplements were even more effective in preventing weight gain and improving overall health. These findings suggest that vitamin D and omega-3 fatty acids could be a promising approach to managing obesity and its complications, not only in animals but potentially in humans as well. This research could lead to better strategies for preventing and treating obesity, benefiting public health.

## 1. Introduction

Obesity, a global public health crisis, affected approximately 18.5% and 14.0% of women and men in 2022, respectively [1]. This issue also impacts pets such as dogs and cats, with their overweight and obesity rates ranging between 19.7 and 59.3% and 11.5 and 52.0%, respectively [2]. Linked to health issues like type II diabetes, osteoarthritis, and cardiovascular diseases, obesity drastically diminishes the quality of life of both humans and pets [3,4,5,6,7].

The high morbidity associated with obesity requires effective prevention measures. Traditional approaches like lifestyle modifications are often unstainable, resulting in only 5–10% weight loss [8]. Pharmacological treatments carry risks, and many drugs have been removed due to their severe side effects [9]. This underlines the need to explore natural products as potential alternatives [10], requiring thorough evaluations to assess the balance between their benefits and risks.

At the macronutritional level, diets limiting sugar intake are viable. For instance, transitioning from a high-fat diet (HFD) to a low-carbohydrate diet in obese mice promoted weight loss and improved glucose metabolism, especially with omega-3 fatty acid (ω3FA) supplementation [11]. A similar diet in obese rats also caused weight reductions [12]. However, there is limited evidence of the long-term sustainability of such diets [13].

Micronutrient interventions could address deficiencies exacerbated by obesity, including vitamin A, B, D, and carotenoid deficiencies [14]. VD benefits obesity management through genomic and non-genomic VD receptor (VDR) activation [15]. Despite contradictory results on weight gain [16,17,18,19], VD positively impacts inflammation [20] and insulin sensitivity [21]. The VD/VDR complex may inhibit C/EBPβ, repressing adipogenesis [22], and regulate fatty acid oxidation and mitochondrial metabolism [23]. VD supplementation in obese rats under tertiary prevention increased antioxidant markers like glutathione peroxidase and superoxide dismutase in adipose tissue [24] and alleviated adipose-tissue-induced inflammation by inhibiting the TNF-α, IL-6, and MCP-1 gene expression in obese mice [25]. VD supplementation has also been shown to improve insulin resistance (IR) in obese, VD-deficient premenopausal women [26]. Emerging evidence suggests that adequate VD levels have a positive impact on the gut microbiota, preventing the progression of metabolic diseases [27]. Indeed, VD administration can interact with the intestinal microbiota, increasing beneficial bacteria such as *Akkermansia* and *Coprococcus* and decreasing the *Firmicutes* phylum and the *Blautia* genus [28].

Another limited nutrient in obesity is omega-3 polyunsaturated fatty acids. These ω3FAs reduced visceral adiposity in HFD rat models [29], partly by activating peroxisome proliferator-activated receptors (PPARs) β/δ, which are involved in fatty acid oxidation [30]. Eicosapentaenoic acid (EPA) and docosahexaenoic acid (DHA) regulate triglyceride levels [31], reduce inflammation [32], and improve insulin sensitivity [33], primarily through GPR120 and/or PPAR activation [32,34]. A ω3FA intervention alleviated dyslipidemia in HFD rat models [35]. EPA inhibits IκB phosphorylation, suppressing the NF-κB signaling pathway and consequently downregulating the expression of inflammation-related genes like TNF-α [36]. ω3FAs increase mitochondrial biogenesis and fatty acid oxidation in rodents, possibly through the activation of PPARα [37]. ω3FA supplementation may promote butyrate-producing bacteria in the intestinal microbiota, contributing to inflammation regulation [38]. Moreover, supplementation in type II diabetic patients improves their insulin sensitivity by reducing non-esterified free fatty acids [39]. Intake of omega-3 is broadly associated with a reduction in the prevalence of obesity and related metabolic diseases in humans [40].

VD and ω3FA co-supplementation could offer a synergistic approach to preventing obesity and its complications. Their complementary mechanisms of action suggest substantial potential benefits. VD, through VDR activation, modulates inflammation, improves insulin sensitivity, and regulates lipid metabolism, while ω3FAs, via PPARβ/δ and GPR120 activation, promote fatty acid oxidation, reduce inflammation, and improve triglyceride regulation. However, the combined effects of these nutrients have seldom been explored in the context of the tertiary prevention of obesity induced by a high-fat, high-sugar (HFHS) diet. Comprehensive research is essential to understand the potential interactions between VD and ω3FAs and to evaluate their combined efficacy in managing obesity and its complications.

Given the roles of VD and ω3FAs in metabolic health, we hypothesize that their use will significantly reduce HFHS-diet-induced obesity and improve the intestinal dysbiosis and metabolic parameters in Wistar rats, with greater effects of the combined intervention. Therefore, this study examines VD and ω3FA supplementation, both individually and in combination, in the tertiary prevention of obesity and its complications in an HFHS-induced obese rat model.

## 2. Materials and Methods

### 2.1. Animal Housing

Sixty-four 8-week-old male Wistar rats (Janvier Labs^®^, Le Genest-Saint-Isle, France) were used after a one-week acclimation. They were healthy, with no genetic modifications, and were not previously involved in any studies. Their environment adhered to the European standard ETS 123, with a 12 h light/dark cycle, a temperature of 22 °C ± 2 °C, and 50% hygrometry.

The procedures followed good practices and the 3R strategy (Replacement, Reduction, Refinement) and were approved by the ethics committee (APAFIS#33784-2021112413036973v3, 13 December 2021) of Pays de la Loire Nantes, France.

### 2.2. Study Design

The rats were divided into two groups (n = 32) and subjected to either a standard diet (S; 3.84 Kcal/g) (3430 Kliba Nafag, Kaiseraugst, Switzerland) or an HFHS diet composed of pellets (4.73 Kcal/g) (D12451 Research Diets^®^, Lynge, Denmark) and sweetened condensed milk (3.22 Kcal/g) (Nestlé^®^, Nantes, France) ad libitum (Figure 1).

Both of these groups were subdivided into four experimental groups each (n = 8) after they had received daily oral supplementations for 13 weeks: the control groups (C) received mineral oil, the VD groups received 600 IU/kg/day of cholecalciferol (NeoBiotech, Xi’an, China), the ω3 groups received 300 mg/kg/day of fish oil (18% EPA/12% DHA) (PhosphoTech Laboratoires^®^, Saint-Herblain, France), and the VD/ω3 groups received both.

At week 0 (W0), week 13 (W13), and week 26 (W26), each rat underwent an oral glucose tolerance test (OGTT), blood sampling, feces collection, and morphometric parameter measurements.

At the end of the supplementation period, the animals were euthanized through an overdose of intraperitoneal anesthetic (pentobarbital, Euthasol^®^, Fort Worth, TX, USA), followed by the ex vivo procedures and organ collection.

These procedures were chosen for their capacity to reliably and reproducibly evaluate the effects of nutritional supplements. The experimental procedures were standardized to minimize potential biases, and the environmental conditions were kept constant to avoid any external influences. All animals were included in the analysis, except those that did not survive to the end of the experiment due to spontaneous death. No additional exclusion criteria were applied.

### 2.3. Weight and Morphometric Measurements

Body weight (BW) and food/water consumption were monitored weekly. Abdominal circumference (AC) and naso-anal length were measured at W0, W13, and W26. Body mass index (BMI) was calculated as follows:BMI (g/cm^2^) = BW (g)/Naso − anal length^2^ (cm^2^)

Feed efficiency (FE) was evaluated as follows [41]:FE (%) = [BW Gain (g)/Caloric Intake (Kcal)] × 100

### 2.4. The Oral Glucose Tolerance Test

At W0, W13, and W26, the OGTT was performed. After a 4 h fast, fasting blood glucose (FBG) was measured using a rodent glucometer (StatStrip Xpress^®^, Nova Biomedical, Waltham, MA, USA) at the end of the tail, and each animal received an oral glucose dose (2 g/kg). Their blood glucose levels were monitored at 15, 30, 45, 60, 90, and 120 min. Given that prolonged fasting in nocturnal animals like rats is increasingly considered suboptimal for metabolic studies, a 4 h fasting period was selected to minimize metabolic and behavioral stress. Extended fasting in rodents can provoke a catabolic state, mobilizing glucose reserves in a way that may interfere with glucose homeostasis assessments and lead to confounding results. A shorter fasting period not only prevents weight loss but also aligns better with the physiological state of the animals, providing stable baseline glycemia for more reliable data [42,43,44].

The area under the curve (AUC) of their blood glucose levels over time was evaluated using GraphPad Prism software (v.9.0.0). The homeostasis model assessment of insulin resistance (HOMA-IR) scores were calculated as follows [45]:HOMA-IR = [Fasting blood insulin (FBI; µIU/mL) × FBG (mmol/L)]/22.5

### 2.5. Blood Sampling

At W0, W13, and W26, after a 4 h fast, the rats were anesthetized with isoflurane (2%), and their blood was collected from their tail tips. EMLA^®^ cream (Lidocaine, Vidal, Paris, France) was applied to prevent pain. The blood was centrifuged (3000× *g*, 4 °C, 10 min) into heparin-containing tubes, followed by plasma aliquoting and storage at −80 °C. The plasma was used for FBI (Laboniris, Nantes, France), leptin (RAB0335, Sigma Aldrich^®^, St. Louis, MO, USA), adiponectin (DY3100-05, Bio-Techne^®^, Minneapolis, MN, USA), and total 25(OH)D (VID21-K01, Eagle Biosciences Inc., Amherst, NH, USA) determination using ELISA kits. Triglycerides (TGs), total cholesterol (TC), and high-density lipoprotein (HDL) were measured by the Nantes hospital center. The triglyceride–glucose (TyG) Index was calculated as follows [46]:TyG Index = ln[TG (mg/dL) × FBG (mg/dL)/2]

### 2.6. Microbiota

Fecal samples were collected at W0, W13, and W26 for the analysis of the intestinal microbiota performed by Biofortis^®^ (Saint-Herblain, France).

DNA extraction:

DNA was extracted using the ZymoBIOMICS™ 96 MagBead DNA Kit (Zymo Research Corp., Tustin, CA, USA), with mechanical and chemical cell lysis. DNA isolation was performed on a KingFisher Flex automated station (ThermoFisher Scientific Inc., Waltham, MA, USA). The DNA was quantified through fluorimetry using a Qubit 3.0 (ThermoFisher Scientific Inc., Waltham, MA, USA).

16S metabarcoding analysis—library preparation and sequencing:

The V3-V4 region of the 16S rRNA gene was amplified through PCR using primers 341F and 785R [47]. The amplicons were cleaned using magnetic AMPure XP beads (Beckman Coulter, Villepinte, France) before adding dual indices and sequencing adapters using the Illumina Nextera XT Index kit (Illumina, San Diego, CA, USA). Each library was cleaned, quantified through fluorimetry (using a Qubit^®^ 2.0 Fluorometer, ThermoFisher Scientific Inc., Waltham, MA, USA), normalized, and pooled. The pooled library was denatured before sequencing (2 × 250 paired-end, v2 chemistry) using an Illumina MiSeq (Illumina, San Diego, CA, USA).

16S metabarcoding analysis—data processing:

The sequences were analyzed using a pipeline developed by Biofortis based on Dadaist2 software [48]. After demultiplexing the barcoded Illumina paired reads, single-read sequences were paired, cleaned, and quality-filtered. Amplicon Sequence Variants (ASVs) were obtained and taxonomically assigned to determine the bacterial community profiles.

### 2.7. Paracellular and Transcellular Intestinal Permeability Assessments

The paracellular and transcellular intestinal permeability was assessed using Ussing chambers. Segments of the proximal duodenum, the distal ileum, the proximal colon, and the distal colon were embedded into sliders (area = 0.4 cm^2^) and mounted onto the Ussing chamber system (Physiologic Instruments, San Diego, CA, USA). The chambers were filled with Krebs solution at 37 °C and supplied with carbogen (95% O_2_; 5% CO_2_). HRP (5.10^−6^ M, P8250-25KU, Sigma Aldrich^®^, St. Louis, MO, USA) and FD4 (10^−4^ M, FD4-1G, Sigma Aldrich^®^) were added to the mucosal side.

Paracellular permeability: Serosal samples were collected every 30 min over 270 min, and the FD4 fluorescence (538 nm) was measured using a TriStar^2^ plate reader (Berthold Technologies^®^, Bad Wildbad, Germany).

Transcellular permeability: The HRP concentration was determined through a colorimetric reaction and read at 450 nm.

For both, the slope of the absorption (ng/mL/min) was calculated as the average absorption over time.

### 2.8. Liver Histology

Following euthanasia, liver samples were immediately fixed in 4% paraformaldehyde and embedded into paraffin, and 3.5 µm thick sections were stained with hematoxylin–eosin–safran (HES). Samples were analyzed by a board-certified veterinary pathologist. Non-alcoholic steatohepatitis (NASH) lesions were scored using a previously described system [49,50], and briefly, the grade of steatosis (from 0 to 3), lobular inflammation (from 0 to 3), hepatocellular ballooning (from 0 to 3), and the grade of fibrosis (from 0 to 4) were evaluated, and a mean score per animal was calculated.

### 2.9. Caecum and Adipose Tissue Analysis

Following euthanasia, the caecum and visceral adipose tissue (VAT)—perirenal, epididymal, and mesenteric fat depots—were collected, blotted dry, and weighed. The adiposity index (AI) was calculated as follows:AI (%) = ∑VAT (g)/BW (g) × 100

The caecum/BW (%) ratio was calculated as follows:Caecum/BW (%) = ∑ [Caecum weight (g)/BW (g)] × 100

### 2.10. Statistical Analyses

The sample size was calculated using BiostaTGV, based on the estimated variation in leptin levels with or without treatment (delta of 35 with a standard deviation of 24 at 5% type I errors and 0.93 power), resulting in 8 animals per experimental group.

Due to the daily interactions between the principal experimenter and the animals, blinding of the experimental in vivo procedures was not feasible. Nevertheless, to mitigate potential bias, histological analyses and blood assays were conducted in a blinded manner.

Data on diet, supplementation, and their interactions were analyzed using Linear Mixed-Effect (LME) models, with a random effect attributed to each rat. The normality and independence of the residuals and random effects were validated according to the recommendations for mixed-effect models [51]. Multiple post hoc comparisons were made using Tukey’s test with adjustment for type I errors [52]. The data were analyzed with R (v.4.3.1) using the nlme, multcomp, plyr, readxl, and tidyverse packages.

Parameters that did not conform to the LME criteria were analyzed using GraphPad Prism (v.9.0.0). The data were assessed for normality using the Shapiro–Wilk test. The analyses were adjusted accordingly for non-parametric data. Comparisons between two groups used Student’s *t*-test or the Mann–Whitney test and an ANOVA or the Kruskal–Wallis test for multiple groups in case of a parametric or non-parametric analysis, respectively.

The significance was set at *p* < 0.05, and results are presented as the mean ± Standard Error of the Mean (SEM), with “n” representing the number of individuals in each group.

## 3. Results

### 3.1. Effects on Morphometric Parameters

Although the BW, BMI, and AC of the HFHS group were lower at W0, these parameters were significantly higher at W13 in the HFHS group, as was the FE, compared to the S group (*p* < 0.001) (Table 1). Moreover, the 13-week HFHS diet induced a significant weight gain, with a 14.85% increase in the HFHS group at W13.

At W26, the HFHS+C, HFHS+VD, and HFHS+VD/ω3 groups had a higher BW, BMI, and AC compared to their respective standard groups, but the HFHS+ω3 group did not. The HFHS+VD/ω3 group showed reduced BW gain compared to its control counterpart (HFHS+C). The BMI values increased more in the HFHS+C group than in the S+C (*p* < 0.01), HFHS+VD (*p* < 0.05), HFHS+ω3 (*p* < 0.001), and HFHS+VD/ω3 (not significant) groups. The HFHS+C group had a marked 12.91% increase in its AC values at W26, which was higher than that in the S+C (*p* < 0.001), HFHS+VD/ω3 (*p* < 0.05), HFHS+ω3 (*p* < 0.001), and HFHS+VD (*p* = 0.063) groups. Although no significant differences in the FE were reported, a trend was noted between the S+C and HFHS+C groups (*p* = 0.059), with a 44.15% higher efficiency with the HFHS diet. All of the supplemented groups normalized their FE. The AI differed significantly among the HFHS groups compared to their standard counterparts. The HFHS+VD group showed a trend towards lower values compared to HFHS+C (*p* = 0.095) (Table 2).

### 3.2. Effects on Glucose Homeostasis

The OGTT showed a 22.42% higher AUC and elevated FBG and FBI levels and HOMA-IR scores in the HFHS group at W13 compared to the S group (*p* < 0.05; *p* < 0.001; and *p* < 0.01, respectively) (Table 3).

At W13, all of the HFHS groups had higher AUCs in the OGTT than their standard matched groups (*p* < 0.001). After the 13-week supplementation, the HFHS+VD, HFHS+ω3, and HFHS+VD/ω3 groups significantly reduced AUCs compared to those of the HFHS+C group (*p* < 0.001) (Table 4).

At W26, only the HFHS+C group maintained higher FBG levels compared to its standard counterpart; it also had higher levels than the HFHS+VD (*p* < 0.05) and HFHS+VD/ω3 (*p* = 0.093) groups. The FBI levels and HOMA-IR scores increased in the HFHS+VD/ω3 group compared to those in its standard counterpart (*p* < 0.01). The HOMA-IR scores tended to be higher in HFHS+C group compared to those in its standard counterpart (*p* = 0.059) (Table 4).

### 3.3. Effects on Lipid Profiles

The 13-week HFHS diet increased TGs by 30.12% (*p* < 0.05), decreased HDL by 15.58% (*p* < 0.05), and increased the TyG Index (*p* < 0.01) (Table 5).

At W26, the HFHS+VD/ω3 group exhibited a significant decrease in its TG levels compared to those in the HFHS+C group (*p* < 0.05). The TyG Index was significantly different between the HFHS+ω3 and HFHS+VD/ω3 groups (Table 6).

### 3.4. Effects on Inflammation

At W0, the leptin or adiponectin levels showed no significant differences between the S and HFHS groups. After 13 weeks, the leptin levels increased significantly in the HFHS group (*p* < 0.001), showing a marked 563% difference in comparison with those in the S group. The adiponectin levels were higher in the HFHS group (*p* < 0.01), although their evolution remained unchanged (Figure 1).

The observed leptin pattern persisted at W26. The evolution of the leptin levels over time showed a lesser increase in leptin levels between the HFHS+C group and both the S+C and HFHS+VD groups. The evolution of leptin was also reduced, but not to the level of significance, in the HFHS+ω3 and HFHS+VD/ω3 groups compared to the HFHS+C group. The adiponectin levels at W26 were higher in the HFHS+VD/ω3 group than its standard counterpart (Table 7).

### 3.5. Plasma Calcidiol Levels

Evaluation of their plasma calcidiol levels showed no significant difference between the S and HFHS groups at W0 or W13 (Figure 2). However, by W26, the groups treated with VD and VD/ω3 demonstrated significant increases in their calcidiol levels. Specifically, the calcidiol levels in the HFHS+VD group increased from W13 to W26, which were statistically significantly different compared to those in the HFHS+C group. In contrast, the HFHS+VD/ω3 group showed a lesser and non-significant increase of 35.13% (Table 8).

### 3.6. Hepatic Steatosis Assessment

The histopathology analysis showed a NASH score of 1.79 ± 0.24 in the HFHS+C group, which was significantly higher compared to the score of 0.00 in its standard counterpart (*p* < 0.001). This score was reduced to 0.36 ± 0.17 by VD supplementation (*p* < 0.001) and to 0.64 ± 0.25 by ω3FA supplementation (*p* < 0.01). The HFHS+VD/ω3 group showed a decrease to 0.92 ± 0.29, although not a significant one, compared to that in the HFHS+C group and remained different from its standard counterpart, which had a score of 0.00 (*p* < 0.05) (Figure 3).

### 3.7. Caecum Morphology

After 26 weeks of the HFHS diet without supplementation, caecum atrophy was observed compared to the condition in the standard counterparts (*p* < 0.01). In the HFHS+ω3 and HFHS+VD/ω3 groups, caecum atrophy was partially reversed, but it was not in the HFHS+VD group (Figure 4).

### 3.8. Intestinal Permeability Assessments

The FD4 assessment revealed reduced paracellular permeability in the duodenum and both colonic segments in the HFHS+ω3 group compared to the HFHS+C group (*p* < 0.05). The HFHS+VD group exhibited reduced the permeability in the duodenum and distal colon sections, although not significantly (Figure 5).

No significant differences in transcellular permeability were observed (Figure 6).

### 3.9. Intestinal Microbiota

At W0, the HFHS group showed higher diversity, but by W13, it had decreased significantly (Figure 7A). At W26, the HFHS+C, HFHS+VD, and HFHS+VD/ω3 groups remained more diverse than their standard groups, and the HFHS+ω3 group showed a slight, but not significant, increase in diversity compared to HFHS+C (Figure 7B).

The Firmicutes-to-Bacteroidetes (F/B) ratio increased significantly by W13 in the HFHS group, although no significant increase was observed for the S group (Figure 8A). By W26, only the HFHS+VD group exhibited a significant difference from its standard counterpart (Figure 8B).

The heatmaps revealed shifts in the microbial populations. In the standard group, there was a reduction in *Prevotella* between W0 and W13 (Figure 9A). Conversely, the HFHS group showed stimulation of the *Akkermansia* population, a significant rise in the *Blautia* genus (exhibiting a 17-fold increase), a decrease in *Alistipes*, the disappearance of *Butyricicoccus*, and the near-complete disappearance of *Prevotella* (reduced by 99.7%), with the emergence of *Lactococcus*. At W26, the HFHS+VD group displayed a 65.8% reduction in *Blautia* compared to the HFHS+C group (Figure 9B). The HFHS+ω3 group showed increases in *Alistipes*, *Duncaniella*, and, notably, *Prevotella*, as well as decreases in *Clostridium sensu stricto* and *Lactococcus* (decreased by 55.1%) compared to the HFHS+C group. The HFHS+VD/ω3 group demonstrated an increase in *Blautia* and a marked 71.9% decrease in *Clostridium sensu stricto* compared to the HFHS+C group. Notably, this latter group was the only group to completely lose its *Sporobacter* population.

## 4. Discussion

This study explored the tertiary prevention of obesity and its complications through the combination of omega-3 fatty acid and vitamin D supplementation in HFHS-diet-induced obesity in Wistar rats. Following a 13-week HFHS diet, which successfully induced obesity, supplementation was administered for another 13 weeks. Our findings support the hypothesis, demonstrating that combined omega-3 and vitamin D supplementation significantly reduced obesity-related complications, including improvements in metabolic parameters and gut health.

Our findings demonstrate that the HFHS diet led to a weight gain of about 15% at W13, classifying the rats as moderately obese [53]. This may have involved a dysregulated energy metabolism, as the obese rats showed an increased FE despite no relative hyperphagia. Intestinal hyperpermeability associated with impairment of the tight junction proteins may be one possibility that could account for the increased FE in diet-induced obesity models [54]. In agreement with this suggestion, previous studies have reported that an increased FE may occur in obese rats without a change in their caloric intake [41,55]. The HFHS diet’s excess calories result in weight gain and fat mass accumulation [56], which is consistent with the observed alterations in morphometric parameters like AC and BMI.

Our study clearly demonstrated an elevated TyG Index in the animals on the 13-week HFHS diet, suggesting IR associated with dyslipidemia. Additionally, glucose metabolism disruption was evident, with increased FBG levels, hyperinsulinemia, and elevated HOMA-IR scores, confirming IR, consistent with similar metabolic alterations previously reported in rats on HFHS diets of various durations, ranging from 6 to 12 weeks [57,58,59,60].

The HFHS diet’s excess caloric absorption led to increased lipogenesis, TG storage, and circulating TG levels, correlating with an increased FE and adipocyte hypertrophy, as previously described [61,62]. Our results demonstrate the coexistence of dyslipidemia and IR, which have frequently been reported to be associated [63]. Although we did not address the mechanism behind these alterations, the overexpression of SREBP-1c, known to activate fat cell differentiation and lipid accumulation [64], may be considered one potential mechanism that could account for the increased TG levels and IR observed in the current study.

Intestinal dysbiosis is a common obesity disorder [65], with high-fat diets significantly altering gut microbiota composition and reducing bacterial richness and diversity. High fat ingestion has been reported to induce strong changes in microbiota composition, affecting the richness and diversity of bacterial species [66]. We observed that the HFHS diet increased the F/B ratio, which is generally ascribed to reduced *Bacteroidetes* and elevated *Firmicutes* [64], a pattern linked to obesity [67,68]. This shift may enhance plasma LPS levels, contributing to endotoxemia [69], as dysbiosis disrupts the intestinal barrier, increasing permeability and LPS translocation via LPS-producing bacteria [70]. These changes in the microbiota likely contribute to HFHS-diet-induced dyslipidemia and IR, as seen in prior diet-induced obesity (DIO) studies [71,72]. *Akkermansia* and *Blautia*, recognized for their roles in reducing inflammation and obesity [73], were found to be more abundant in our DIO model, which matches the findings of other studies [74,75,76] and may reflect an adaptive response to metabolic disturbances. Their beneficial effects, supported by previous studies [77,78], include producing butyrate and deoxycholic acid, which combat obesity-related inflammation. *Lactococcus*, another key genus, possesses probiotic and anti-inflammatory properties [79], though its elevated abundance in high-fat diets has been associated with increased leptin levels [76]. Conversely, several genera, including *Alistipes* and *Butyricicoccus*, are negatively correlated with obesity markers like BMI, triglycerides, and fasting blood glucose, with the latter also producing beneficial butyrate [80,81,82]. *Prevotella*, linked to low-fat, high-fiber diets, presents a complex relationship with obesity, with studies reporting conflicting changes in its abundance [75,83,84], possibly due to the genetic diversity within the genus [85]. These findings underscore the intricate interplay between the gut microbiota and metabolic health in obesity.

In our study, the obesity and obesity-related disorders observed at W13 persisted until W26. This included elevated BW, BMI, AC, inflammation, and intestinal dysbiosis, highlighting the long-term impact of the HFHS diet on metabolic health. We revealed a significant increase in VAT in the HFHS+C group, substantiated by a high AI. This is in line with the statement that VAT is more biologically and metabolically active than subcutaneous adipose tissue [86].

Our study demonstrates that VD supplementation significantly improves several metabolic and physiological parameters under an HFHS diet, which seems to be correlated with calcidiol levels. Although the HFHS diet did not induce VD deficiency in the supplemented rats, consistent with previous research showing no change in serum 25(OH)D levels but higher 1,25(OH)_2_D levels in obese mice [87], the VD supplementation still provided notable benefits. This aligns with the literature indicating that obesity alters VD metabolism, with elevated PTH and disrupted regulation of 1,25(OH)_2_D due to changes in VD-metabolizing enzymes [88]. Thus, VD supplementation enhances calcidiol status and mitigates some metabolic disturbances associated with obesity, despite the absence of vitamin D deficiency. The VD levels observed in our study, approximately 120 ng/mL in the vitamin-D-supplemented rats, slightly exceed the reference range of 20–100 ng/mL but remain well below the intoxication threshold of >150 ng/mL [89]. In our study, VD supplementation did not lead to a significant reduction in weight gain compared to that in the HFHS+C group but did result in a slight reduction in abdominal circumference and fat accumulation. VD was found to normalize FE, suggesting improved metabolic efficiency or reduced lipid absorption. The improvement in FE and the reduction in abdominal circumference indicate that VD may help limit fat accumulation and indirectly enhance glucose homeostasis. These findings are consistent with the existing literature linking VD deficiency to increased adiposity and impaired glucose metabolism [90,91]. The improvement in glucose homeostasis and the reduction in abdominal circumference observed in our work further support VD’s role in mitigating obesity-related metabolic disturbances.

VD positively influences gut and liver health in DIO models, prompting our investigation into whether its anti-obesity effects involve gut- and liver-dependent mechanisms. VD supplementation slightly reduced intestinal permeability, suggesting enhanced gut barrier integrity, which may have contributed to a reduced abdominal circumference by limiting lipid absorption. This aligns with findings in VDR^−^ mice showing VD’s role in maintaining intestinal barrier integrity [92]. Additionally, the HFHS+VD group exhibited a blunted leptin increase, indicating potential improvements in leptin sensitivity, likely mediated by VD’s anti-inflammatory effects, which include downregulating pro-inflammatory cytokines (TNF-α, IL-6) and inhibiting NADPH oxidase [93,94,95,96]. However, VD did not significantly alter the F/B ratio, suggesting its benefits are not linked to alleviating gut dysbiosis. Beyond its intestinal effects, VD supplementation reduced hepatic steatosis, pointing to a potential protective mechanism via hepatic VDR activation, which may alleviate NASH. The observed reduction in triglyceride levels supports this hypothesis, as VD likely enhances lipid degradation by promoting lipolysis and fatty acid β-oxidation through PPAR-α signaling, consistent with prior research [97,98].

Our study also aimed to investigate whether ω3FA supplementation could prevent obesity-related metabolic disturbances, focusing on its effects on BMI, AC, and FE. ω3FA supplementation did not lead to a significant reduction in overall weight gain compared to that in the HFHS+C group. However, it significantly limited increases in BMI and abdominal circumference and tended to normalize the FE. These results suggest that ω3FA can reduce central obesity and improve metabolic efficiency, consistent with findings that ω3FA can impact adiposity and energy expenditure [99,100]. Furthermore, the activation of GPR120 by ω3FA induces thermogenesis, enhancing energy expenditure [99]. Additionally, we observed improved glucose tolerance and a slight normalization of FBG levels, indicating enhanced glucose homeostasis and insulin sensitivity. This aligns with the literature showing ω3FA’s role in improving insulin sensitivity and metabolic function [101].

ω3FA supplementation also led to reduced hepatic steatosis, suggesting that ω3FA improves liver health by reducing fat accumulation and enhancing lipid metabolism. This finding is supported by evidence that ω3FA can enhance hepatic lipid oxidation and inhibit lipogenesis [99,100] and aligns with findings that an 8-week fish oil treatment in HFD mice could reduce hepatic steatosis [102].

Similarly to VD, the ω3FA supplementation improved the metabolic parameters in the rats fed the HFHS diet. ω3FA reduced intestinal permeability, supporting gut barrier integrity and potentially reducing systemic inflammation. While it did not significantly alter the Firmicutes-to-Bacteroidetes (F/B) ratio, ω3FA positively affected the gut microbiota by increasing beneficial genera such as *Alistipes* and *Prevotella* and decreasing *Clostridium sensu stricto*. These changes suggest that ω3FA helps manage obesity-related disturbances by modulating the gut microbiota. Additionally, flaxseed oil, rich in alpha-linolenic acid, has been shown to reduce colonic damage in DSS-induced colitis by improving oxidative status and modulating inflammatory factors while partially restoring the integrity of the intestinal epithelial barrier [103]. This further underscores the potential of ω3FA to influence gut health and reduce inflammation through multiple mechanisms.

VD/ω3 co-supplementation showed greater improvements in certain parameters compared to individual supplementation. It significantly reduced BW gain, decreased the TG levels at week 26, and notably lowered the abundance of *Clostridium sensu stricto* compared to the HFHS+C group. The reduction in *Clostridium sensu stricto*, linked to obesity and elevated plasma leptin levels [76,80], suggests a gut-flora-mediated mechanism in limiting BW gain. The additive effects of co-supplementation on BW gain and TG reductions may have resulted from the complementary actions of VD and ω3FA, both known to enhance lipolysis and fatty acid β-oxidation [97,98,99]. However, co-supplementation did not outperform individual supplementation in improving the other metabolic parameters, such as FE or glucose tolerance. The mild to moderate benefits observed in the weight and TG reductions may not have been sufficient to counteract broader obesity-related disturbances. Potential nutrient–nutrient interactions at the kinetic level between VD and ω3 may have limited their combined effectiveness to only specific parameters. Further studies are needed to clarify these interactions and their impact on metabolic health.

To understand the effects of VD/ω3 co-supplementation better, we measured the plasma 25(OH)D levels at week 26 in the HFHS+VD and HFHS+VD/ω3 groups. The plasma 25(OH)D levels increased more significantly in the HFHS+VD group, while the HFHS+VD/ω3 group showed only a smaller, non-significant rise. This suggests that ω3FA, at the dose studied, negatively modulates plasma VD levels, potentially limiting the combined supplementation’s effectiveness compared to that of the individual treatments. The underlying mechanisms behind the reduced plasma VD levels in the HFHS+VD/ω3 group were not directly examined but may involve impaired metabolism or defects in VD’s absorption. Supporting this, studies have shown that ω3FA co-supplementation reduces the ratio of 25(OH)D to its catabolite 24,25(OH)D, indicating altered VD metabolism [104]. Additionally, ω3FA may limit VD’s absorption by forming mixed micelles in the intestinal lumen, further affecting VD’s bioavailability [105,106].

This study has several limitations. The dose and duration of VD and ω3FA supplementation may not have been sufficient to detect subtle or long-term effects. A pilot study would have been relevant to optimizing these parameters. Additionally, we did not perform a detailed analysis of the physicochemical properties of the VD/fish oil mixture (i.e., chemical composition, stability, and interactions), which may have affected the outcomes by potentially altering the bioavailability or efficacy of the supplements. Future research should explore higher doses and longer supplementation periods to determine whether these adjustments would result in greater benefits. Moreover, a comprehensive analysis of the chemical composition, stability, and interactions within the VD/ω3 mixture is essential to understand their potential additive or antagonistic effects and to ensure the reliability and reproducibility of our experimental outcomes.

## 5. Conclusions

In conclusion, this study highlights the significant benefits of long-term supplementation with VD and ω3FA, alone and in combination, in managing obesity and its comorbidities in HFHS-diet-induced obesity in Wistar rats. Our results confirmed that 13- and 26-week HFHS diets induced obesity in rats, characterized by dyslipidemia, impaired glucose regulation, intestinal dysbiosis, and altered leptin levels. While both individual supplementations of VD and ω3FA had beneficial effects, co-supplementation was particularly effective in preventing weight gain, reducing TGs, and decreasing *Clostridium sensu stricto*, which is associated with obesity and leptin production (Figure 2).

Thus, we believe that long-term VD/ω3 co-supplementation may have implications for the tertiary prevention of obesity and related disorders. Further studies are needed to explore the cellular mechanisms and kinetic interactions between VD and ω3FA to optimize their use in the framework of nutritional interventions targeting experimental DIO.

## Data Availability

The data that support the findings of this study are freely available at the following link: https://filesender.renater.fr/?s = download&token = a0061fdd-b607-4bb1-bc02-cfa281a4224a (accessed on 12 November 2024).

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
