# Peer review of "Omega-3 Fatty Acid and Vitamin D Supplementations Partially Reversed Metabolic Disorders and Restored Gut Microbiota in Obese Wistar Rats"

_biology, 2024, doi:10.3390/biology13121070_

Round 1
Reviewer 1 Report (Previous Reviewer 5)
Comments and Suggestions for Authors
All my previous comments are addressed in this revised version, and I don't have any further suggestions. This revised version could be accepted for publication.
Author Response
Please see the attachment, "Reviewer 1"

Reviewer 2 Report (New Reviewer)
Comments and Suggestions for Authors
The manuscript is dedicated to a current and modern topic - the scientific substantiation of the consumption of vitamin B3 and omega-3 fatty acids for the prevention and treatment of obesity. The relevance of the study is adequately described. The choice of research methods is reasonable, the size of the experimental groups seems sufficient. The results clearly describe the benefits of vitamin B3 and omega-3 fatty acid supplementation in a rat model of obesity. The results are discussed for all groups.
The main comment is the gross inconsistency between the title and the content of the article. This is because the title talks about 'Improved disorders', whereas the results talk about 'Reducing' obesity-related pathology.
Another comment in section 2.4 (lines 152-159). This text would look better in the Discussion section.
Otherwise there are no comments.
Author Response
Please see the attachment, "Reviewer 2".

Reviewer 3 Report (New Reviewer)
Comments and Suggestions for Authors
In the manuscript “Omega-3 Fatty Acids and Vitamin D Supplementations Improved Metabolic Disorders and Gut Microbiota in Obese Wistar Rats”, the authors have described the differential and combinatorial effect of vitamin D and Omega 3 fatty acid. in the diet-induced rats’ model. They have discussed about the dietary interventions of them to mitigate Obesity-related metabolic disorder with various statistical analyses. I would like to say that this is an extensive study with adequate data provided by authors to prove their hypothesis, but the manuscript is lacking the mechanism of action how vitamin D and omega 3 fatty acid mitigate obesity and gut dysbiosis. I have raised few comments and suggestions while evaluating the manuscript. It’s a minor revision.
1. Is there any reason why the authors have particularly chosen Vitamin D and Omega 3 fatty acid for the present study? Please clarify.
2. How did the authors choose the concentration of Vitamin D and Omega 3 fatty acid for this study? Are these daily requirements or FDA-approved daily supplementation? Please clarify.
3. The discussion section is vast. I suggest the authors reduce the unnecessary discussion part from this section if possible.
4. I suggest the authors include microphotography of liver histopathology along with NASH score in the Figure 3 or else provide them as a supplemental figure.
5. Though the authors provided too much of data to support their hypothesis. I would like to suggest the authors provide one small schematic diagram for your experimental design and grouping to understand easily.
6. Please check general English grammar and typographical errors throughout the manuscript.
Author Response
Please see the attachment, "Reviewer 3".

Round 2
Reviewer 2 Report (New Reviewer)
Comments and Suggestions for Authors
Article is ready to publish
This manuscript is a resubmission of an earlier submission. The following is a list of the peer review reports and author responses from that submission.
Round 1
Reviewer 1 Report
Comments and Suggestions for Authors
This research value is very low considering this combination of vit D and Omega 3 has been tested on humans for most of the parameters shown in this study.
1) You mentioned in Figure 5. that at W0 the HFHS group showed higher diversity but by W0 13 it increased significantly but, that is not true and is a red flag for the whole study. W0 is showing that HFHS group has already started off the study with a significant different diversity index for the gut microbiome than the S (standard diet group). This indicates that the groups random selection wasn't followed appropriately. Figure 6 is also showing a considerable higher Firmicutes/ Bacteroidetes in the W0 for HFHS to begin with so, the follow up increase in the W13 can be biased.
2) Tables 2,4,6,8 and 10 only shows W0 and W13 means for the groups but not week 26, why? same for the graphs that only shows W0 and W13 but not week 26. Even with the tables and graphs that show W26 individual parameters you should still compare the means for that week with W0 and W13!
3) Some of the abbreviation I couldn't find what it is for e.g. AI? A list of all the abbreviation would be very helpful.
4) For glucose tolerance or fasting test. The standard fasting period is 12-16 hours not 4 hours? Why you only did fasting for 4 hours?
Author Response
Please see the attachment, section "Response to Reviewer 1 Comments".

Reviewer 2 Report
Comments and Suggestions for Authors
Dear authors, an interestinga nd good paper. But a point can be improved in my opinion. The 3.9 Intestinal microbiota can be improved, because the gut microbiota is descibed strain by strain, but with lack of physiological or health related benefits dealing with teh other results and specifically the gut permeabililty. In details line 405 to 416 derseve a partial conclusion. What are the implications of the microbial shifts you present ?
Author Response
Please see the attachment, section "Response to Reviewer 2 Comments".

Reviewer 3 Report
Comments and Suggestions for Authors
The impact of supplementation with omega-3 fatty acids and vitamin D for weight reduction is generally known. Supplementation with vitamin D and omega-3 fatty acids can affect faster fat loss and support weight loss processes, as long as we suspect deficiencies in these components. This is because they are extremely important for the overall health of our body. Therefore, I believe that the issue addressed in the publication entitled: Omega-3 Fatty Acids and Vitamin D Supplements Improved Metabolic Disorders and Gut Microbiota in Obese Wistar Rats is very relevant to the fight against one of the diseases of civilization - obesity.
The research presented here was planned in a clear and comprehensive manner and carried out extensively.
The discussion of the results is comprehensive and supported by the scientific literature. The authors suggested what is a weakness of the paper (lines 572-582) and should be checked in the next stage of the research.
I believe that the paper meets the criteria and is suitable for publication in the special issue of Physiology and Pathophysiology of Obesity of the journal Biology.
Author Response
Please see the attachment, section "Response to Reviewer 3 Comments".

Reviewer 4 Report
Comments and Suggestions for Authors
The work of Jan et al. is quite interesting and contains a lot of information, however corrections are needed before publication. The introduction needs to be connected a little better, but there is enough necessary information to understand the issue. The results need to be presented better and more clearly, while the discussion needs to be expanded because it is too general and the data is not well connected. There is no good sequence in the discussion. I will also list minor corrections that need to be redone.
In the summary, more precisely in line 24, it is unnecessary to write the control, given that it is known that the control must always exist.
Line 42 - I suggest putting more recent data. Now we are in the year 2024, so it would be better to put a percentage for the year 2024 or at least 2020, since the number of obese people in the world is even higher. According to data from 2024, nearly 880 million adults and 159 million children and adolescents aged 5-19 years are living with obesity.
Line 71 -the above text and this one should be better connected. This way there is no sequence, it is not a continuation, but like a thrown section. It should be written more fluidly and connected between omega-3 and VD.
2.1. Animal Housing - date of approval is missing
2.2. ad libitum and ex vivo - Latin names should be written in italics.
Line 140 - write how glucose was measured.
Line 184 – correct writing the chemical symbols (should be O2).
2.10. Statistical Analyses - Write p value in italics. Please revise it in the entire paper.
Vs also should be written in italics. Please revise it in the entire paper.
Line 312 - marked 563% difference- please check and correct this information.
Table 7 – It is better to show the graphic representation of leptin as image a and adiponectin as image b.
It would also be better to display table 9 graphically.
Line 346 - Histopathology analysis showed a NASH score of xx in the HFHS+C group – XX? the number is missing.
Check figure 5a and written. HFHS groups had lower diversity at week 26 compared to standard-fed animals.
The discussion is insufficient considering the amount of data. Also, additional explanations are missing. The discussion would have been better if all the groups had been commented on together with regard to the measured parameters and a comparison of explanations had been made.
Line 477 is redundant because the whole sentence is repeated and continued in the next line.
References are not written according to the journal's instructions. Please edit.
Comments on the Quality of English LanguageEnglish is fine.
Author Response
Please see the attachment, section "Response to Reviewer 4 Comments".

Reviewer 5 Report
Comments and Suggestions for Authors
Dear authors,
The research article “Omega-3 Fatty Acids and Vitamin D Supplementations Improved Metabolic Disorders and Gut Microbiota in Obese Wistar Rats” by Jan et al. is interesting. I believe readers would benefit from it. However, the authors may consider making necessary amendments to the manuscript for better comprehensibility of the study. Having said that, your original submission has some potential for improvement, and some of the issues need to be addressed, for which I recommend major revision. I've included a few of them below:
1. There is text overlapping of 20%, please reduce it.
Specifically, please rewrite 2.6 Microbiota section (most of this section was used from https://www.mdpi.com/2076-2607/12/6/1098).
2. It would be beneficial for the readers to have a schematic presentation/summary presentation of the key points of the study and highlight VD and ω3FA, alone or combined, offer significant benefits in preventing obesity, gut dysbiosis, and metabolic alterations, with the VD/ω3combination showing the most promise.
3. Please elaborate introduction and/or discussion section with some important references linking oxidative stress and, intestinal dysbiosis, and inflammation, gut microbiome influenced by several factors, including obesity, are missing in other species as well, including murine model, for example: https://pubmed.ncbi.nlm.nih.gov/37701897/
https://www.ncbi.nlm.nih.gov/pmc/articles/PMC7561009/
https://www.ncbi.nlm.nih.gov/pmc/articles/PMC9866669/
https://www.mdpi.com/2076-2607/12/9/1831
https://www.frontiersin.org/journals/nutrition/articles/10.3389/fnut.2021.781622/full
4. Any results on pro-inflammatory markers (hs-CRP, IL-6, and TNF-α) and one anti-inflammatory cytokine (IL-10)?
Author Response
Please see the attachment, section "Response to Reviewer 5 Comments".

Round 2
Reviewer 1 Report
Comments and Suggestions for Authors
Comments 1: You mentioned in Figure 5. that at W0 the HFHS group showed higher diversity but by W0 13 it increased significantly but, that is not true and is a red flag for the whole study. W0 is showing that HFHS group has already started off the study with a significant different diversity index for the gut microbiome than the S (standard diet group). This indicates that the groups random selection wasn't followed appropriately. Figure 6 is also showing a considerable higher Firmicutes/ Bacteroidetes in the W0 for HFHS to begin with so, the follow up increase in the W13 can be biased. |
Response 1: Thank you for your insightful comment, which we have carefully considered. We have corrected in the manuscript by replacing "increased" with "decreased" in line 398 to accurately reflect the data. Regarding the random selection, the animals were indeed randomized into two groups. However, as is often the case with randomization, there may have been an unintended imbalance in baseline characteristics, such as the gut microbiota diversity. Although the difference in the Firmicutes/Bacteroidetes ratio at W0 between the HFHS and Standard groups was observed, but it did not reach statistical significance. Nonetheless, the increase observed at W13 in the HFHS group was much more pronounced than in the Standard diet group. To clarify this point further, we have added the following statement to line 410: ", although no significant increase was observed for the S group." Response: Figure 6 which is now figure 8 in the new version, the error bars of W0 HFHS doesn’t seem to be equal (inside one is shorter) with that said, it is mentioned in the paper that the difference between the S group and HFSH group is not significant for W0. By glancing at it with those tight error bars of the S group it seems to be significantly different. In contrast to figure 5 or 7 in the new version with those wide error bars it seems unlikely for them to be significantly different!! I have the same reservation on the new figure 1B W13!! |
Comments 4: For glucose tolerance or fasting test. The standard fasting period is 12-16 hours not 4 hours? Why you only did fasting for 4 hours? |
Response 4: The standard fasting period of 12-16 hours is not unanimously supported and not scientifically justified (https://doi.org/10.1152/ajpendo.90617.2008). 12h-16h period, the equivalent of overnight fasting, in contrast to Humans, is less appropriate for exploring glucose homeostasis in small rodents. Moreover, it was reported that mice lost more weight after 16 h of fasting compared with 4 h (https://doi.org/10.1177/0023677213501659). The prolonged (overnight) fasting provokes a catabolic state which mobilizes the glucose reservoirs that will affect their metabolism and the subsequent assessment by IPGTT (10.2147/DMSO.S234665). Shorter fasting times are now considered more physiological to reduce metabolic stress in rodents. There are recommendations for glucose tolerance testing for mice, and these should be similarly applied to rats (10.1242/dmm.006239; https://norecopa.no/media/6351/food-deprivation.pdf). In our study we have chosen a shorter fasting period (4h) to reduce metabolic stress and weight loss. Some studies have shown that gastric content is reduced as much after 4–6 h of fasting as after 12 or 18 h (10.1016/j.vascn.2008.05.115). Fasting more than 4h prior to investigate glucose homeostasis has been proposed to be not physiological and should be proscribed (10.1016/j.molmet.2020.101058). Response: You should reference those in the paper itself in the material and method section for this test. Your own reference 10.1016/j.molmet.2020.101058 stated that fasting glucose levels doesn’t significantly change before 6h of fasting. According to the material and method you are measuring glucose tolerance; that is your main parameter so, you should have chosen the fasting period protocol that ensures proper data acquisition. They have also mentioned that Insulin tolerance was comparable between baseline and 6 h fasts. So, you should have at least gone with 6h fasting which was also the recommendation in this recent report, Guidelines and Considerations for Metabolic Tolerance Tests in Mice 2020, https://doi.org/10.2147/DMSO.S234665”. They also stated that 12-16h protocol can be applied taking the proper steps to protect the mice. |
Comments 2: Tables 2,4,6,8 and 10 only shows W0 and W13 means for the groups but not week 26, why? same for the graphs that only shows W0 and W13 but not week 26. Even with the tables and graphs that show W26 individual parameters you should still compare the means for that week with W0 and W13! |
Response 2: Thank you for this observation. The rationale behind our presentation is that Tables 1, 3, 5, 7, and 9 display the effects of the diet on the parameters studied in the absence of supplementation. In contrast, Tables 2, 4, 6, 8, and 10 focus on the effects of the supplements (VD, ω3, VD/ω3) provided from week 13 to week 26. In these latter tables, week 13 is considered the baseline because it reflects the values before supplementation was introduced. This presentation allows a clear comparison of the impact of supplementation from the W13 to W26 without any confusion with the diet-dependent effect. Response: I would have agreed with that if the parameters compared in the two situations were the same but you are comparing the aggregations of the two groups on W0 and W13 then on W13 and W26 you are comparing an array of parameters that I have no reference of how they looked like before the high fat diet feeding and the subsequent treatment. Considering that these two groups were significantly in some parameters to start with before any food changes or treatments make those comparisons critical. Also, the statistical analysis seems off. The letters that state the significance seems to be placed on the standard deviation not the actual mean or average value which is very confusing to assess. It would have been much better to have the mean value and standard deviation on the same line with smaller font than having them dispersed. Also, in table 4 it seems that you comparing the AUC of W13 with FBG of W13 for group S+VD and showing that as significant (have the same letter a)??These are totally different parameters and I don’t see a point of comparing them, I would understand that you would report a significance between AUC of W13 and W26 or FBG of W13 and W26. You could see this statistic significance mix up in a lot of the parameters. Also, I am not sure but some of the numbers that are showing significance doesn’t make since to me. For example, also in table 4, you have in S+C group FBG W13 97.9+/-3.2 vs W26 95.7+/-3.5 being significantly different from each other? 97.9-3.2= 94.7 that is lower than/same value range of 95.7 mean/average of W26?
|
Reviewer 5 Report
Comments and Suggestions for Authors
All my comments are addressed. However, some of the statistical analyses are missing. Please include statistical analyses for Figure 2, Figure 5b, Figure 6, and Table 6.
